# Systemic Treatment in Intermediate Stage (Barcelona Clinic Liver Cancer-B) Hepatocellular Carcinoma

**DOI:** 10.3390/cancers16010051

**Published:** 2023-12-21

**Authors:** Dimitrios S. Karagiannakis

**Affiliations:** Academic Department of Gastroenterology, Laiko General Hospital, Medical School of National and Kapodistrian University of Athens, 12462 Athens, Greece; dkarag@med.uoa.gr

**Keywords:** hepatocellular carcinoma, liver cancer, intermediate stage, BCLC-B, systemic treatment, tyrosine kinase inhibitors, immune checkpoint inhibitors

## Abstract

**Simple Summary:**

According to the latest BCLC algorithm, the reference treatment of BCLC-B HCCs that are non-eligible for surgery or liver transplantation is trans-arterial chemoembolization. However, some BCLC-B HCCs are unsuitable for (i.e., those with diffuse, infiltrative, and extensive bilobular disease) or refractory to this treatment. To date, it has not been clarified whether systemic treatment plays a role in these cases. In this review, we provide recent data regarding this issue. Interestingly, it was shown that systemic treatment might benefit patients with BCLC-B2 and B3 hepatocellular carcinoma. Moreover, a combination of two systemic agents or even a triple combination with trans-arterial chemoembolization has been tried, giving promising results. However, further validation by RCTs is necessary before moving towards a modification of the current treatment guidelines.

**Abstract:**

Hepatocellular carcinoma (HCC) represents an entity of poor prognosis, especially in cases of delayed diagnosis. According to the Barcelona Clinic Liver Cancer (BCLC) staging system, patients in BCLC-A are the most suitable for potentially curative treatments (surgery or radiofrequency ablation), whereas those in BCLC-C should be treated only with systemic treatment, as locoregional interventions are ineffective due to the tumor’s extensiveness. For patients in the BCLC-B stage, trans-arterial chemoembolization (TACE) is the reference treatment, but the role of systemic treatment has been constantly increasing. As this group of patients is extremely heterogeneous, a case-by-case therapeutic strategy instead of a one-fits-all treatment is certainly required to achieve adequate results against HCC. The decision of selecting among immune checkpoint inhibitors (ICIs), tyrosine kinase inhibitors (TKIs), TACE, or a combination of them depends on the patient’s tumor load, the severity of liver dysfunction, the general performance status, and the presence of concomitant extrahepatic diseases. The objective of this review is to critically appraise the recent data regarding the systemic treatment of BCLC-B HCCs, aiming to emphasize its potential role in the management of these difficult-to-treat patients.

## 1. Introduction

Liver cancer constitutes one of the most frequent cancers worldwide, accounting for approximately 905,000 new cases yearly [1]. Hepatocellular carcinoma (HCC) is the most common type of liver cancer, representing almost 90% of cases, followed by cholangiocarcinoma [2]. The vast majority of HCC (80–85%) is developed in the background of chronic liver disease, mainly cirrhosis, and its early diagnosis is a great challenge for hepatologists and oncologists [3]. 

The early diagnosis of HCC is of great importance, as it is associated with better prognosis [4]. Thus, all the international guidelines propose HCC surveillance programs in high-risk patients by using liver ultrasound (US) every 6 months, with or without the concurrent use of alpha-fetoprotein (AFP) [5,6,7]. Nevertheless, the majority of HCCs are still diagnosed with a delay when they have already reached more advanced stages [8]. 

The delay in HCC diagnosis greatly restricts the options of treatment, as eradicating procedures such as liver resection (LR) or radiofrequency/microwave ablation (RFA/MWA) seem not to be beneficial at the advanced stages of HCC. Furthermore, even in cases of early diagnosis, when potentially curative interventions with high 5-year or even 10-year survival rates can be applied, most patients exhibit high (up to 70%) 5-year recurrence rates. In order to increase recurrence-free survival, the induction of systemic treatment earlier in the course of disease has been supported by many investigators if not for all patients at least for those at higher risk, as defined by the presence of microvascular invasion, satellite nodules, or a proliferative histological pattern [9,10,11,12,13,14]. Compared to other solid tumors, the number of systemic agents against HCC has been traditionally small. Until recently, multiple-target tyrosine kinase inhibitors (TKIs) were the only choice for patients with advanced HCC. Sorafenib was the first systemic agent to be approved for HCC treatment, followed by regorafenib, lenvatinib, and cabozantinib [15,16,17,18]. These agents were shown to prolong the overall survival (OS) but, unfortunately, by no longer than a few months. Moreover, they often cause moderate or severe adverse events (AEs), resulting in dose reduction or early interruption [5]. Recently, the systemic treatment landscape significantly changed after the confirmation that the HCC microenvironment is potentially susceptible to immunotherapy (IO). This led to the introduction of immune checkpoint inhibitors (ICIs) as an alternative and very challenging therapeutic option in patients with HCC. However, monotherapy with ICIs failed to improve the OS despite durable response rates of approximately 15–20% [19,20,21]. Thus, the combination treatment of an ICI with an anti-angiogenic factor or the co-administration of two ICIs was endorsed. Undoubtedly, the combination of atezolizumab, which is an ICI of the programmed cell death receptor-1 (PD-L1), with bevacizumab, an antibody that inhibits the vascular endothelial growth factor (VEGF) ligand A, provided better median survival compared to sorafenib (19.2 vs. 13.4 months, respectively), with minimal grade 3 to 4 AEs [22]. Moreover, positive results were also obtained from the combination treatment of tremelimumab, a cytotoxic T-lymphocyte antigen-4 (CTLA-4) inhibitor, with durvalumab, a PD-L1 inhibitor. This combination offered a longer median OS and higher 3-year survival rate compared to sorafenib [23]. Based on the aforementioned good results, the latest BCLC algorithm recommended atezolizumab/bevacizumab or tremelimumab/durvalumab as the first treatment option in advanced BCLC-C stage patients, followed by TKIs as second-line agents [24]. To date, except for BCLC-C HCCs (portal invasion and/or extrahepatic spread, preserved liver function, and performance status (PS) 1–2, the use of systemic treatment has also been extended to those BCLC-B HCCs (multinodular HCC, preserved liver function, and PS 0) that have been characterized as unsuitable for trans-arterial chemoembolization (TACE) (i.e., those with diffuse, infiltrative, and extensive bilobular disease). 

It has been investigated whether systemic treatment could also have a place in BCLC-B patients with less extensive neoplastic disease. It has been established that some patients suitable for TACE BCLC-B HCCs do not respond sufficiently to this therapeutic intervention [25]. In these cases, recurrent TACE sessions have been found incapable of improving survival, while they have been associated with a higher risk of procedure-related liver failure and death [26]. The modified KINKI criteria have classified BCLC-B HCCs into B1, B2, and B3 substages. Compared to B1, where TACE seems to be effective, B2 and B3 are commonly refractory to TACE, and an earlier induction of systemic therapy instead of controversial repeated TACE sessions could be more beneficial [27]. In addition, the early induction of systemic treatment before liver function deteriorates has been associated with better OS and PFS [26]. 

This review aims to identify whether systemic treatment has a role in the treatment of BCLC-B HCCs. Specifically, based on recent data, we herein try to elucidate which BCLC-B patients could be the best candidates for systemic treatment and, highlight what could be the optimal initiation time. Furthermore, we analyzed data on combination treatments with two or more factors (among ICIs, TKIs, and TACE) to illustrate potential therapeutic alternatives for these difficult-to-treat patients. It has to be emphasized that the role of systemic treatment as a neoadjuvant procedure for HCC downstaging was out of the scope of this review. 

## 2. Tyrosine Kinase Inhibitors (TKIs)

TKIs have multiple anti-tumor effects and are widely used in several types of cancers. They down-regulate different molecular pathways that take part in carcinogenesis. The primary targets are the tyrosine kinase receptors (RTKs), key proteins that regulate cancer growth and metastasis [28,29]. Specifically, TKIs block the phosphorylation of tyrosine kinases and the subsequent signaling pathways, slowing down cancer growth [30]. Some of the inhibited networks are the rat sarcoma (RAS)/mitogen-activated protein kinases (MAPKs), phosphoinositide 3-kinase (Pi3K)/protein kinase B (AKT)/mechanistic target of rapamycin (mTOR), phospholipase C (PLC)/Ca2+/calmodulin-dependent protein kinase-protein kinase C (CaMK-PKC), Janus kinase (JAK)/signal transducer and activator of transcription protein family (STAT), epidermal growth factor receptor (EGFR), vascular endothelial growth factor receptor (VEGFR), fibroblast growth factor receptor (FGFR), platelet-derived growth factor receptor (PDGFR), hepatocyte growth factor receptor (HGFR, Met), and RAF kinases [30,31]. Due to their multiple actions, TKIs were the standard treatment of care in advanced HCC over the last 15 years, but after the induction of IO, they now comprise the second-line option.

### 2.1. Sorafenib

Sorafenib is a small molecular inhibitor of several RTK pathways, including VEGFR 1–3, RAF (CRAF, BRAF, mutant BRAF), PDGFR (PDGFR-β), and MAPK [32]. Its approval as a first-line systemic treatment against advanced, unresectable HCC was received from the FDA in 2008 [15] and EMA in 2009 after the results of the SHARP trial, a multicenter, phase 3, double-blind, randomized, sorafenib- versus placebo-controlled clinical trial [32]. The study included patients with PS less than or equal to 2, compensated liver disease Child–Pugh (C-P) A, adequate hematologic and renal function, and a life expectancy of at least 12 weeks. Subjects with viral and non-viral liver disease were equally distributed in both groups, and the vast majority of them were categorized in the BCLC-C group (82% in the sorafenib group and 83% in the placebo group versus 18% and 17% of BCLC-B, respectively, in the two groups). The study met its primary endpoint, as the median OS was significantly higher in sorafenib compared to the placebo group (10.7 vs. 7.9 months; HR: 0.69; 95% CI: 0.55–0.87, *p* < 0.001), representing a 31% lower risk of death. The study also showed a significantly prolonged time to radiological progression in the sorafenib group vs. placebo (5.5 vs. 2.8 months; HR: 0.58; 95% CI: 0.45–0.74, *p* < 0.001), while 2% of patients in the former group experienced partial response versus 1% in the latter. Regarding safety profile, the overall incidence of treatment-related AEs was 80% in sorafenib compared to 50% in the placebo, with the most common sorafenib-induced AEs being gastrointestinal, constitutional, or dermatologic (grades 1 and 2), while diarrhea was the most common grade 3 AE (8% in the sorafenib group). Discontinuation rates due to AEs were similar between the two groups (close to 38%). 

Afterward, in 2016, the GIDEON study prospectively showed a significant difference in OS between C-P A and C-P B sorafenib-treated patients (13.6 vs. 5.2 months, respectively), without significant differences in type and severity of AE [33]. In line with that, McNamara et al. conducted a meta-analysis of 30 studies, with 8678 C-P A/B patients receiving sorafenib as a first-line systemic treatment for advanced HCC [34]. The authors demonstrated that treatment discontinuation rates without HCC progression as well as treatment-related deaths did not significantly differ between C-P A and B patients. However, C-P B stage was an independent factor of poor prognosis (4.6 months survival in C-P B vs. 8.8 in CP-A, *p* < 0.0001). The results from this meta-analysis as well as those from the GIDEON study illustrated a worse prognosis in cases of treatment initiation after the development of liver decompensation. Unfortunately, a sub-analysis investigating differences in OS between BCLC-B and BCLC-C patients was not performed in any of the aforementioned studies. 

Ogasawara S et al. in 2014 and Arizumi T et al. in 2015 first evaluated the efficacy of sorafenib in BCLC-B hepatocellular carcinoma as being refractory to TACE [35,36]. The former group of investigators retrospectively reviewed 509 patients treated with TACE, 122 of whom had refractory HCC. After excluding patients with a C-P score of ≥8 (patients with C-P C and C-P B/8 were excluded) and/or advanced-stage BCLC-C, 20 out of 122 patients converted to sorafenib, and 36 continued with TACE. Interestingly, the time to disease progression (TTP), defined as the time towards the development of C-P C or advanced BCLC-C stage, was 22.3 months in the conversion group and 7.7 months in the non-conversion group (*p* = 0.001), while the OS was significantly higher in the former compared to the latter group (25.4 vs. 11.5 months, respectively, *p* = 0.003) [36]. Likewise, Arizumi et al., among 56 non-responders to TACE patients, retrospectively found a median OS of 24.7 months in those who switched to sorafenib (n = 32) compared to 13.6 months (*p* = 0.002) in those who continued with TACE (n = 24) [36]. The above evidence demonstrated the necessity of shifting from TACE to systemic treatment without delay in cases where TACE was ineffective. 

Later on, Ren et al. compared the efficacy of TACE versus sorafenib plus TACE in patients with unresectable HCC. A total of 308 patients (247 in TACE monotherapy and 61 in TACE/sorafenib) were retrospectively evaluated. In the overall analysis including all patients, the median OS was significantly longer in the combination group than in the TACE monotherapy group (29 ± 7.2 vs. 14.9 ± 1.1 months; *p* = 0.008). In the propensity matching cohort (61 subjects receiving TACE/sorafenib and 122 receiving only TACE), the median OS was 29 ± 7.2 months in the combination group and 14.9 ± 1.5 in the TACE group (HR 0.684, 95% CI: 0.470–0.997; *p* = 0.018). Exclusively in BCLC-B subjects, the median OS after matching was 33 ± 9.8 months in the combination arm and 25.3 ± 6.7 months in the arm of TACE monotherapy (HR 0.620, 95% CI: 0.345–1.114; *p* = 0.041) [37]. These findings were in opposition to the results provided by Meyer et al. The latter group of investigators, in a randomized, placebo-controlled, double-blind, phase 3 trial, randomly assigned 313 C-P A patients with unresectable HCC to TACE plus sorafenib (n = 157) or TACE plus a placebo (n = 156). Interestingly, no significant difference was determined between the sorafenib and the placebo group regarding the PFS (238 days (95% CI: 221–281) vs. 235 days (95% CI: 209–322), respectively) (HR 0.99; 95% CI: 0.77–1.27, *p* = 0.94) [38]. Recently, Kudo et al. established the superiority of TACE/sorafenib compared to TACE alone in a randomized, open-label, multicenter, prospective research that included a large proportion of BCLC-B HCCs (n = 44/80 (55%) in the first group and n = 34/76 (45%) in the second) as well as a significant number of BCLC-A cases (33.8% and 43.4%, respectively; patients with single tumor but unresectable due to size > 5 cm). Although TACE plus sorafenib did not show significant OS benefit over TACE alone (median OS 36.2 months in TACE/sorafenib vs. 30.8 months in TACE monotherapy; HR 0.861; 95% CI: 0.607–1.223; *p* = 0.40), the combination offered a clinically meaningful prolongation of OS (ΔOS 5.4 months) and a significantly improved PFS (22.8 vs. 13.5 months in TACE/sorafenib vs. TACE alone, respectively) (HR 0.661; 95% CI: 0.466–0.938; *p* = 0.02) [39]. Additionally, time to vascular invasion, time to extrahepatic spread, and time to stage progression were significantly longer in the combination group, whereas a post hoc analysis revealed PFS and OS benefits in HCC patients with tumor burden beyond the up-to-seven criteria.

### 2.2. Regorafenib

The RESORCE trial demonstrated the role of regorafenib in HCC treatment. The primary endpoint of OS in the treatment group was favorable, with an HR of 0.63 relative to the placebo (*p* < 0.0001). Specifically, regorafenib compared to the placebo significantly extended the median OS (10.6 vs. 7.8 months, respectively), whereas it prolonged the PFS by 1.6 months (3.1 vs. 1.5 months, respectively). Additionally, TTP was 3.2 months in regorafenib and 1.5 months in the placebo group (HR: 0.44; 95% CI: 0.36–0.55, *p* < 0.001), while the disease control rate (DCR) and overall response rate (ORR) were 65.2% and 10.6% in regorafenib and 36.1% and 4.1% in the placebo, respectively. AEs were reported in all 374 regorafenib recipients (100%) and 179/193 (93%) of placebo recipients. In this trial, 14% of patients in the regorafenib group and 11% in the placebo group had BCLC-B HCC. However, a sub-analysis merely in BCLC-B was not performed, probably due to the small number of included patients [16].

Later on, the REFINE study recruited 500 patients with HCC, of whom 482 (97%) had been previously treated with sorafenib. Regorafenib was the second-line treatment in 81% of patients (n = 403), third-line or higher in 17% (n = 87), and first-line in only 2% (n = 8). The investigators evaluated the OS in patients who had previously received sorafenib, taking into account the C-P stage and the albumin–bilirubin (ALBI) grade at the entry. The median OS was 16 months in C-P A versus 8 months in C-P B subjects. The median OS among those with ALBI grades 1, 2, and 3 was 19.6 months (95% CI, 14.8–19.6), 10.5 (95% CI, 8.7–16.0), and 3.1 months (95% CI, 1.6–8.7), respectively. These results underlined the importance of switching to second-line treatments without delay when the first-line treatment is found to be ineffective, and furthermore, they showed that the better the liver function at the time of conversion, the better the survival [40]. Subsequently, several studies evaluated the effect of regorafenib on OS, PFS, or ORR [41,42,43] but without estimating the effect of regorafenib in patients with BCLC-B HCCs exclusively. Recently, Han Y et al., in a retrospective real-world study, tried to explore the benefits and tolerability of TACE combined with regorafenib in patients with unresectable HCC who had failed in the first-line treatment with sorafenib. Eighteen BCLC-B and twenty BCLC-C patients were included. After a median follow-up of 5.6 months (range: 0.7–17), the median OS, PFS, and TTP were 14.3, 9.1, and 9.1 months, respectively. The PFS and TTP were found to be associated with AFP levels, tumor size, dose of regorafenib, and degree of response (complete response (CR) vs. partial response (PR) vs. stable disease (SD)), while the OS was associated only with regorafenib’s dose and degree of response. The BCLC stage was not found to correlate with OS, PFS, and TTP in the Cox regression analysis [44]. Afterwards, Lee et al., in a retrospective study of 108 patients (BCLC stage B/C: 18.5%/81.5%; albumin–bilirubin (ALBI) grade 1/2/3: 40.7%/58.3%/0.9%; C-P A/B: 84.3%/15.7%) with unresectable HCC treated with regorafenib after sorafenib failure, found no significant difference in PFS between BCLC-B and BCLC-C stages; however, the ALBI grading was associated with OS (2–3/1: HR 2.758, 95% CI: 1.458–5.216, *p* = 0.002) and post-progression survival (ALBI 2/1: HR 4.499, 95% CI: 1.541–13.137, *p* = 0.006 and 3/1: HR 26.926, 95% CI: 6.638–109.227, *p* < 0.001) [42]. Obviously, the study did not detect any improvement in PFS by using regorafenib earlier regarding the BCLC stage, but it clarified an association between the severity of liver dysfunction at the time of regorafenib induction and patients’ outcome.

### 2.3. Lenvatinib

Lenvatinib is an orally acting antiangiogenic agent that inhibits VEGFR 1–3, FGFR 1–4, PDGFR-α, and RET and KIT tyrosine kinases. In a phase 3 trial (REFLECT), lenvatinib, as a treatment of unresectable HCC, was found to be non-inferior to sorafenib in terms of OS. The trial demonstrated the clinical significance of lenvatinib over sorafenib regarding PFS, TTP, and ORR [17]. In 2019, Kudo et al. conducted a proof-of-concept study to compare lenvatinib versus TACE to explore whether the former is more favorable for intermediate-stage HCC with large or multinodular tumors exceeding the up-to-seven criteria. Among 176 eligible subjects (unresectable HCC, beyond the up-to-seven criteria, no prior TACE/systemic therapy, no vascular invasion, no extrahepatic spread, and C-P A), 30 were treated with lenvatinib and 60 with TACE after propensity score matching for patients’ demographics. Change in ALBI score from the baseline to the end of treatment was from −2.61 to −2.61 in lenvatinib patients (*p* = 0.254) and from −2.66 to −2.09 in TACE patients (*p* < 0.01), respectively. The lenvatinib group showed significantly higher ORR (73.3% vs. 33.3%; *p* < 0.001), significantly longer median PFS (16 vs. 3 months; *p* < 0.001), and significantly longer OS compared to TACE (37.9 vs. 21.3 months; HR: 0.48, *p* < 0.01) [45].

Subsequently, Kobayashi et al. prospectively investigated whether treatment with lenvatinib improved survival in TACE-naïve patients with BCLC substage B2 HCC and preserved liver function. Thirty-one patients were enrolled, and substage B2 was defined as C-P score 5–6, beyond the up-to-seven criteria, PS 0, and no portal vein thrombosis. By reviewing the studies published to date, the authors concluded that the median OS and the 1-year survival rates for substage B2 patients treated with TACE were between 15.6 and 26.9 months and 59.2 and 75.5%, respectively [46,47,48,49,50,51]. Given values as historical controls, the threshold of 1-year survival rate after TACE was set to 60%, and this was expected to further increase to 78% after treatment with lenvatinib instead of TACE in patients with substage B2 disease. According to the results, the authors found a median OS of 17.0 months and 1-year survival rate of 71.0%. The 2-year survival rate was 32.3%, the median PFS was 10.4 months, and the 1-year PFS rate was 42%. In addition, according to the Response Evaluation Criteria in Solid Tumors 1.1 (RECIST 1.1), an ORR of 22.6% and a DCR of 90.3% were revealed, which were 70% and 90.3%, according to the modified RECIST (m-RECIST), respectively [52]. The overall lenvatinib safety profile was comparable to that observed in the REFLECT trial [17]. The authors performed a sub-group analysis to determine the prognostic factors in lenvatinib-treated patients, taking into account the age (≥ or <75 years), the AFP level (≥ or <400 ng/mL), the tumor diameter (≥ or <50 mm), the number of liver tumors (≥ or <10), and the C-P score (5 or 6). Interestingly, the C-P score was found to affect the prognosis, with a median OS of 21.5 months in C-P 5 compared to 13.2 months in C-P 6 (HR: 3.206; 95% CI: 1.081–9.509). Obviously, lenvatinib was found to be more effective in treatment-naïve BCLC B2 substage HCCs compared to TACE, especially in patients with C-P A/5 [52].

The significance of BCLC and C-P stage was also identified in the study of Patwala et al. In a real-world Australian multicenter cohort of 155 patients (BCLC stage C (69.7%) and BCLC stage B (27.7%); C-P A/B/C: 78.8%/19.7%/1.5%), the authors retrospectively showed that patients treated with lenvatinib had median OS and PFS of 7.7 months and 5.3 months, respectively. In Kaplan–Meier analysis, improved OS was noticed in patients who had developed hypertension or diarrhea or had proceeded in dose reduction compared to patients who had not presented the above AEs (median OS: 16.2 vs. 9.4 months, *p* = 0.02; 17.5 vs. 10.1 months, *p* = 0.08; 19.6 vs. 7.8 months, *p* < 0.01, respectively). Conversely, patients with more severe liver disease were associated with worse OS (C-P B/C vs. C-P A: median OS 5.6 vs. 12.5 months; *p* < 0.01). The development of AEs was attributed to a more potent action of lenvatinib, which positively affected the OS, while on the other hand, the advanced C-P stage (B/C) resulted in worse outcome due to less hepatic reserve. Significantly, the multivariate analysis demonstrated that the BCLC stage (HR: 2.50, 95% CI: 1.40–4.45, *p* < 0.01), the baseline albumin (HR: 0.89, 95% CI: 0.86–0.93, *p* < 0.01), the development of hypertension (HR: 0.42, 95% CI: 0.24–0.73, *p* < 0.01) or diarrhea (HR: 0.47, 95% CI: 0.25–0.88, *p* = 0.01), and the dose reduction (HR: 0.41, 95% CI: 0.24–0.69, *p* < 0.01) were independently associated with OS. Similarly, dose reduction (HR: 0.45, 95% CI: 0.29–0.68, *p* < 0.01), older age (HR: 0.96, 95% CI: 0.94–0.98, *p* < 0.01), and higher baseline C-P score (HR: 1.24, 95% CI: 1.01–1.52, *p* = 0.04) were independent predictors of PFS [53]. Clearly, the above results indicated that lower C-P score and milder BCLC stage at the beginning of treatment both predispose towards a better prognosis.

The effectiveness of lenvatinib’s earlier induction was also confirmed by the research of Hiraoka et al. This was a multicenter, retrospective study that aimed to determine whether the cause of liver cirrhosis (NAFLD vs. non-NAFLD) had any additional impact on the OS and PFS in lenvatinib-treated HCC patients. Interestingly, the Cox regression analysis revealed that elevated ALT (≥30 U/L) (HR 1.247, *p* = 0.029), modified ALBI grade 2b (HR 1.236, *p* = 0.047), and elevated AFP (≥400 ng/mL) (HR 1.294, *p* = 0.014) and NAFLD (HR 0.763, *p* = 0.036) were significant prognostic factors of PFS. Furthermore, higher AFP (≥400 ng/mL) (HR 1.402, *p* = 0.009), BCLC-C stage (HR 1.297, *p* = 0.035), later introduction of lenvatinib (HR 0.737, *p* = 0.014), and modified ALBI grade 2b (HR 1.875, *p* < 0.001) were independently associated with OS [54].

Except for lenvatinib’s efficacy as monotherapy in patients with BCLC-B HCC, some investigators explored the role of lenvatinib plus TACE as a combination treatment. Hence, Ando et al. collected 88 BCLC-B subjects previously controlled by taking lenvatinib and divided them into two groups. Thirty patients, who continued with lenvatinib plus TACE comprised the first group, and fifty-eight patients who remained on lenvatinib monotherapy comprised the second. After matching, the two groups had similar characteristics (BCLC stage B (100%); beyond up-to-seven criteria (68.4% vs. 63.16%); ALBI grade 2 (31.58% vs. 31.58%)). No significant difference in ORR was found between the two groups (63.2% vs. 63.2%). However, the multivariate analysis identified that the TACE (HR: 0.264, 95% CI: 0.087–0.802, *p* = 0.019) and C-P score 5 (HR: 0.223, 95% CI: 0.070–0.704, *p* = 0.011) were independent significant predictors of PFS when added. The median PFS was 11.6 months in the first group and 10.1 months in the second, while CR was detected in 15.8% of the former and 10.5% of the latter group. The survival rate was significantly higher in the lenvatinib–TACE group compared to the lenvatinib group (median survival time; not reached vs. 16.9 months, *p* = 0.007). Furthermore, lenvatinib-associated AEs were equally presented in two groups [55]. Almost simultaneously, a retrospective study was conducted to evaluate whether the addition of lenvatinib in TACE-treated patients improved prognosis in comparison to TACE alone. The 1-year and 2-year OS findings were significantly higher in TACE + lenvatinib (88.4% and 79.8%) than TACE alone (79.2% and 49.2%, *p* = 0.047). Similarly, the former group had better PFS rates (1-year PFS rate: 78.4% vs. 64.7%; 2-year PFS rate: 45.5% vs. 38.0%, *p* < 0.001, respectively). The ORR was also higher in the TACE/lenvatinib group (ORR: 68.3% vs. 31.7%, *p* < 0.001). Interestingly, there were no significantly different rates of hepatic deterioration (increase in C-P stage) between the two groups. When the analysis was limited only to BCLC-B patients, the combination group showed better ORR (69.7% vs. 38.5%, *p* = 0.016), higher PFS (HR: 0.149; 95% CI: 0.059–0.379, *p* < 0.001), and a trend toward higher OS (HR: 0.28; 95% CI: 0.092–0.853, *p* = 0.07) [56].

### 2.4. Cabozantinib

Cabozantinib is an oral multi-kinase inhibitor that inhibits the activity of VEGF, cMET, RET, AXL, TIE2, and FLT3. The survival-prolonging effects of cabozantinib, as a second-line agent for patients with HCC refractory/intolerant to sorafenib treatment, compared to the placebo control were presented at the CELESTIAL trial. This was a double-blind, phase 3 trial that randomized patients with HCC refractory to prior sorafenib treatment in a 2:1 ratio of cabozantinib 60 mg/day versus the matching placebo. Patients had C-P A liver function and PS 0–1. Among 707 subjects, 470 were assigned to cabozantinib and 237 to the placebo. Most of them had BCLC-C HCC (91% and 90% in the two groups, respectively). At the end of the trial, cabozantinib had significantly improved the OS, PFS, and ORR compared to the placebo. The median OS was 10.2 months in the cabozantinib arm versus 8 months in the placebo arm (HR 0.76; 95% CI: 0.63–0.92; *p* = 0.005), with a median PFS of 5.2 versus 1.9 months, respectively (HR 0.44; 95% CI: 0.36–0.52; *p* < 0.001). In the cabozantinib arm, the ORR was 4%, and the DCR was 64%, while the placebo arm offered an ORR of <1% and a DCR of 33%. AEs of any grade were reported in 99% of patients treated with cabozantinib and in 92% of those treated with the placebo, with rates of 68% and 36% for grade 3/4 AE, respectively [18]. Of note, a separate analysis of cabozantinib’s efficacy exclusively on BCLC-B HCC was not conducted, probably due to the small number of participants at that stage. Likewise, several studies that followed the CELESTIAL trial did not evaluate the potential role of cabozantinib in patients with BCLC-B HCC exclusively [41,57,58,59]. 

Regarding the combination of cabozantinib with ICIs, a multicenter, open-label, randomized, phase 3 trial (COSMIC-312) enrolled 837 individuals with HCC not amenable to curative or locoregional therapy and not previously treated with any kind of systemic treatment. These patients were randomly assigned to a combination treatment of cabozantinib plus atezolizumab (n = 432), sorafenib monotherapy (n = 217), or cabozantinib monotherapy (n = 188). At the end of the follow-up (median period of 15.8 months), the PFS was more significantly prolonged in the combination treatment than in sorafenib monotherapy (6.8 vs. 4.2 months, respectively; HR 0.63; 99% CI: 0.44–0.91, *p* = 0.0012), but the OS (interim analysis) did not differ between the two groups (15.4 vs. 15.5 months, respectively; HR 0.90; 95% CI: 0.69–1.18, *p* = 0.44). Importantly, a sub-analysis exclusively on BCLC-B patients was not performed, even though these patients comprised 30% of each group [60]. Thus, the efficacy of cabozantinib/atezolizumab combination treatment in BCLC-B stage was not evaluated. 

### 2.5. Immune Checkpoint Inhibitors (ICIs)

The vast majority of studies evaluating the efficacy of ICIs combined with anti-VEGFR monoclonal antibodies or TKIs have been carried out in populations predominantly consisting of BCLC-C patients [61]. In 2022, Hiraoka et al. retrospectively analyzed the results from 171 HCC patients treated with atezolizumab plus bevacizumab. The study included an adequate number of BCLC-B HCCs (n = 68, 40%) and showed an ORR and a DCR after 6 weeks of 10.6% and 79.6%, respectively. A similar response was observed in patients with or without a history of systemic treatment. In 111 patients who underwent a 6-week observation period, the ALBI score was significantly worsened 3 weeks after treatment introduction (−2.525 ± 0.419 vs. −2.323 ± 0.445, *p* < 0.001) but recovered at week 6 (−2.403 ± 0.452; *p* = 0.001 for comparison to ALBI score at week 3). Nonetheless, the study did not provide comparative data regarding the ORR and DCR between BCLC-B and BCLC-C patients [62]. One year later, the same group of investigators compared the efficacy of atezolizumab/bevacizumab versus lenvatinib as a first treatment for unresectable HCC (BCLC-B/C) and did not show significant differences in the response rates of two groups by using the m-RECIST criteria. Furthermore, the PFS and OS were comparable between the two groups, but after adjusting with inverse probability weighting, the atezolizumab/bevacizumab group showed better PFS (0.5/1/1.5 years: 56.6%/31.6%/non-estimable vs. 48.6%/20.4%/11.2%, *p* < 0.0001) and OS (0.5/1/1.5 years: 89.6%/67.2%/58.1% vs. 77.8%/66.2%/52.7%, *p* = 0.002) [63]. Concurrently, the impact of the combination of ICIs with TACE was assessed. Yang et al. compared the efficacy and safety of regorafenib plus ICIs and TACE (R+ICIs+TACE) versus regorafenib plus ICIs (R+ICIs) as a second-line treatment for patients with HCC. After propensity score matching, patients who received R+ICIs+TACE had higher ORR (34.8% vs. 4.3%, *p* = 0.009) and longer PFS (5.8 vs. 2.6 months, *p* = 0.014) compared to those who received R+ICIs. The Cox regression model showed that age (>50 vs. ≤50 years old) (HR 0.260, 95% CI: 0.112–0.605, *p* = 0.002), C-P stage (A6+B7 vs. A5) (HR 3.125, 95% CI: 1.293–7.555, *p* = 0.011), and treatment option (R+ICIs+TACE vs. R+ICIs) (HR 0.244, 95% CI: 0.096–0.622, *p* = 0.003) were independent prognostic factors of PFS, while AFP (>400 vs. ≤400 ng/mL) (HR 2.625, 95% CI: 1.194–5.770, *p* = 0.016), platelets/lymphocytes ratio (high vs. low) (HR 2.384, 95% CI: 1.006–5.648, *p* = 0.048), ALT (≤35 vs. >35 U/L) (HR 0.405, 95% CI: 0.176–0.932, *p* = 0.034), and treatment option (R+ICIs+TACE vs. R+ICIs) (HR 0.410, 95% CI: 0.170–0.988, *p* = 0.047) were independent prognostic factors of OS. On the contrary, the BCLC stage was not found to correlate with the PFS or OS [64]. 

Likewise, Liu et al. conducted a meta-analysis of five studies and 425 subjects to compare the efficacy and AEs between TACE plus TKIs plus ICIs versus TACE plus TKIs. It was clarified that the first option improved the ORR (RR 1.53, 95% CI: 1.27–1.85, *p* < 0.01) and extended the PFS (mean difference: 4.51 months, *p* < 0.01) and OS (mean difference: 5.75 months, *p* < 0.01). The fixed-effects model showed a significant difference in OS among C-P A patients considering the treatment option (TACE/TKIs/ICIs vs. TACE/TKIs: 22.6 vs. 15.1 months, *p* = 0.05). Similarly, the OS was significantly higher in TACE/TKIs/ICIs than TACE/TKIs among C-P B subjects (12.9 vs. 6.5 months, respectively, *p* < 0.01) [65].

Two ongoing studies are testing the efficacy and safety of atezolizumab in combination with bevacizumab compared to TACE in patients with intermediate-stage liver cancer. The NCT04803994 trial has been designed to evaluate the efficacy and safety of atezolizumab/bevacizumab plus TACE versus TACE alone, while the DEMAND trial (EUDRACT 2019–002430-36), a randomized, two-arm, non-comparative, phase II study, aims to assess the efficacy of atezolizumab/bevacizumab followed by on-demand selective TACE as well as the efficacy of the initial synchronous treatment of TACE plus atezolizumab/bevacizumab in patients with unresectable HCC.

## 3. Discussion

According to the latest BCLC algorithm, systemic treatment is the standard of care for patients with BCLC-C HCCs [24]. Whereas systemic treatment was initially recommended only to BCLC-C HCCs, it is consequently implemented also in BCLC-B after the confirmative results of poor response to TACE in some BCLC-B cases [36]. Eventually, it was verified that BCLC-B HCCs do not consist of a homogeneous group, and therefore, a one-for-all treatment is not supported [66]. According to the modified KINKI criteria, BCLC-B patients are stratified to three sub-groups based on the C-P score and the degree of tumor dispersion (total size and number of lesions), while different therapeutic interventions are recommended for each of the three sub-groups [67]. Specifically, in BCLC-B3 (C-P score 8,9 and any tumor status), systemic treatment rather than TACE is the therapeutic option of choice, as TACE has been found to be inadequate in controlling the disease and improving the prognosis. On the contrary, TACE consists of a suitable option for BCLC-B1 and B2 stages [61]. However, in recent studies, a positive impact of systemic treatment on BCLC-B2 patients (C-P score 5–7 and beyond up-to-7 tumor burden) has emerged, raising concerns about which should be the optimal therapeutic decision for such patients. According to the studies of Kudo et al. and Kobayashi et al., lenvatinib might offer better ORR and PFS compared to TACE in BCLC-B2 stages [45,52]. Albeit impressive, these results were not extracted from randomized, controlled (RCT), two-arm, lenvatinib versus TACE studies, and therefore, further validation is needed. Nevertheless, BCLC-B patients non-responsive to TACE should switch to lenvatinib as soon as possible and surely before liver function deteriorates to C-P B/ALBI grade 2 or HCC aggravates to BCLC-C stage [54,55,56]. Concerning the potential positive effect of adding lenvatinib to TACE in BCLC-B stages, it has been demonstrated that combination treatment is probably superior to TACE or lenvatinib monotherapy, but this issue needs to be clarified by RCTs [55,56]. 

Sorafenib studies have also verified that the early switch from TACE to sorafenib in cases of refractoriness to TACE may prolong the TTP and OS in BCLC-B subjects [35,36]. Furthermore, the combination of TACE plus sorafenib has been associated with higher OS compared to TACE monotherapy in naïve individuals with unresectable HCC. Importantly, this superiority has been found not only when combination treatment has been applied to the entire population but also in BCLC-B patients in particular. However, these results were drawn from retrospective analysis, whereas RCTs provide conflicting evidence regarding this issue [37]. Thus, Meyer et al. affirmed that combination treatment does not improve PFS compared to TACE monotherapy, while on the other hand, Kudo et al. documented that combination prolongs the PFS and post-progression survival (PPS). The latter group of investigators demonstrated that combination treatment may offer better tumor control, giving the chance for subsequent post-protocol therapies [38,39]. 

Concerning regorafenib, relatively few studies have evaluated its role in BCLC-B HCCs. Similar to sorafenib or lenvatinib studies, it has been shown once again that, after failure to sorafenib, regorafenib must be initiated as soon as possible, as the lower the C-P score and the ALBI score at the time of the switch, the better the results. An early switch from sorafenib to regorafenib is suggested not only when regorafenib is used as monotherapy but also when it is combined with TACE [40,44]. Regarding cabozantinib, data about its efficacy on BCLC-B HCCs in particular are lacking. 

Apparently, when combination treatments of atezolizumab/bevacizumab and tremelimumab/durvalumab were introduced in the therapeutic array of HCC, the natural course of the disease changed [22,68]. Furthermore, investigations into novel IO agents are underway, and several combinations of two or three therapeutic components have been evaluated. The aspect of dual combination (ICIs with TKIs or TACE) or triple combination (ICIs/TKIs/TACE) is another issue of interest. It has been speculated that the addition of TACE boosts the response to immunological agents by causing tumor-specific antigen release, while ischemic cell damage might increase VEGF levels, increasing the effectiveness of anti-angiogenic agents [45]. Considering these potential benefits, investigators conducted studies to evaluate the role of triple-combination treatments in HCC. Interestingly, triple therapy was found to be superior to dual combination with respect to ORR, PFS, and OS. Moreover, this superiority was confirmed not only among patients with C-P A but also among C-P B subjects [64,65]. However, as the above promising results were extracted from retrospective studies and not from RCTs, further validation is required before recommending triple therapy in BCLC B/C patients. 

The objective of this review was neither to illustrate the whole sum of published studies regarding HCC treatment nor to highlight all the ongoing trials on novel agents but to provide data regarding the potential role of systemic treatment in BCLC-B HCCs (Table 1 and Table 2). The discussion of other therapeutic options for BCLC-B HCCs such as surgery, liver transplantation, or use of systemic treatment as neoadjuvant procedure for HCC downstaging is out of the scope of this review. Thus, considering the aforementioned studies, it is becoming clear that BCLC-B1 HCCs may be treated with TACE if they are not suitable for liver transplantation [66]. In case of insufficient response to TACE, treatment must rapidly change to ICIs or TKIs (lenvatinib or sorafenib if ICIs are not feasible). The lower the C-P score at the time of the switch, the higher the benefits. Subjects with BCLC-B2 HCCs may be treated with TACE or ICIs depending on the size and number of neoplastic lesions. Several studies comparing TACE to ICIs in BCLC-B2 are underway. In specific cases (i.e., one large and several small nodules), the combination of TACE/ICIs might be supported. If ICIs are not feasible, sorafenib or lenvatinib might be used as an alternative. The combination of lenvatinib/TACE might be an option in specific cases (i.e., in patients with one large and several small nodules), while the triple combination of ICIs/TACE/TKIs might be used with caution in certain patients with adequate liver function. When TACE is applied as first-line treatment and is found insufficient to mitigate tumor load, then a rapid shift to ICIs or TKIs (lenvatinib or sorafenib if ICIs are not feasible) must be considered. Importantly, the transition must be carried out as soon as possible, definitely before liver function deteriorates. Individuals with BCLC-B3 HCC should be treated according to the latest BCLC criteria (ICIs as the first treatment, followed by TKIs or TKIs from the beginning if ICIs are not feasible). 

## 4. Conclusions

In conclusion, it seems that patients with BCLC-B2 and BCLC-B3 HCCs that are refractory or intolerant to TACE may benefit from systemic treatment (Figure 1). New therapeutic plans have been tried, and potential combinations have emerged. However, the promising results of recent studies must be further validated by RCTs before moving on to modification of the current treatment guidelines.

## Figures and Tables

**Figure 1 cancers-16-00051-f001:**
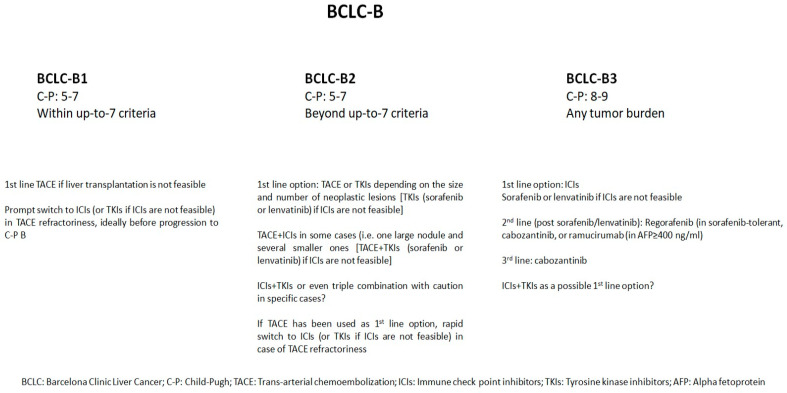
Suggested treatment options concerning BCLC-B sub-stages.

**Table 1 cancers-16-00051-t001:** Efficacy of TKIs on patients with intermediate (BCLC-B) HCC.

*Tyrosine Kinase Inhibitors*			
*Sorafenib*			
Authors/Year/Ref No.	Study	Description	Results
Llovet J.M. et al. (2008) [32]	RCT	Sorafenib vs. placebo C-P A, PS 0–1 BCLC-C (80% in each group)	***Sorafenib superior to placebo*** Median OS: 10.7 vs. 7.9 months; HR: 0.69; 95% CI: 0.55–0.87, *p* < 0.001. TTP: 5.5 vs. 2.8 months; HR: 0.58; 95% CI: 0.45–0.74, *p* < 0.001. AE: 80% vs. 50%.
Marrero J.A. et al. (2016) [33]	Observational	Sorafenib in C-P A vs. sorafenib in C-P B	***Better OS in C-P A*** OS: 13.6 vs. 5.2 months. No significant differences in the type and severity of AE.
McNamara M.G. et al. (2018) [34]	Meta-analysis	30 studies, 8678 C-P A/B patients in the sorafenib group as a first-line systemic treatment for advanced HCC	***Better OS in C-P A*** Treatment discontinuation rates without HCC progression and treatment-related deaths without significant difference between C-P A and C-P B. OS: 4.6 months in C-P B vs. 8.8 in C-P A, *p* < 0.0001.
Ogasawara S. et al. (2014) [35]	Retrospective	122 patients refractory to TACE 20 converted to SOR vs. 36 continued with TACE	***Better results after switching to sorafenib*** TTP: 22.3 vs. 7.7 months (*p* = 0.001). OS: 25.4 vs. 11.5 months (*p* = 0.003).
Arizumi T. et al. (2015) [36]	Retrospective	56 patients non-responders to TACE 32 switched to SOR vs. 24 continued with TACE	***Better results after switching to sorafenib*** OS: 24.7 vs. 13.6 months (*p* = 0.002).
Ren B. et al. (2019) [37]	Retrospective	SOR+TACE vs. TACE	***SOR/TACE superior to TACE in the entire population as well as in BCLC-B in particular*** OS: 29 ± 7.2 vs. 14.9 ± 1.1 months; *p* = 0.008). In the propensity matching cohort (61 patients in SOR/TACE vs. 122 patients in TACE → median OS: 29 ± 7.2 vs. 14.9 ± 1.5 months (HR 0.684, 95% CI: 0.470–0.997; *p* = 0.018). Exclusively in BCLC-B → median OS: 33 ± 9.8 vs. 25.3 ± 6.7 months (HR 0.620, 95% CI: 0.345–1.114; *p* = 0.041).
Meyer T. et al. (2017) [38]	RCT	SOR/TACE (n = 157) vs. TACE (n = 156)	***No difference between SOR/TACE and TACE alone*** PFS: 238 vs. 235 days (HR 0.99; 95% CI: 0.77–1.27, *p* = 0.94).
Kudo M. et al. (2022) [39]	RCT	SOR/TACE vs. TACE BCLC-B: n = 44/80 (55%) in the first group and n = 34/76 (45%) in the second	***SOR/TACE superior to TACE*** Median OS: 36.2 vs. 30.8 months (HR 0.861; 95% CI: 0.607–1.223; *p* = 0.40). The SOR/TACE offered a clinically meaningful prolongation of OS (ΔOS 5.4 months). PFS: 22.8 vs. 13.5 months (HR 0.661; 95% CI: 0.466–0.938; *p* = 0.02).
** *Regorafenib* **			
Bruix J. et al. (2017) [16]	RCT	REG vs. placebo	***REG superior to placebo*** OS: 10.6 vs. 7.8 months (HR: 0.63, *p* < 0.0001). PFS: 3.1 vs. 1.5 months. TTP: 3.2 vs. 1.5 months (HR: 0.44; 95% CI: 0.36–0.55, *p* < 0.001). AE: in 100% of regorafenib recipients and 93% of placebo recipients. No sub-analysis merely in BCLC-B.
Merle P. et al. (2020) [40]	Observational	500 patients with HCC of whom 482 patients had been previously treated with SOR Subsequent switch to REG	***Better results in an early switch from Sorafenib to Regorafenib*** Evaluation considering liver function at the time of the switch (C-P A vs. C-P B and ALBI-1 vs. ALBI-2 vs. ALBI-3). Median OS: 16 months in C-P A versus 8 months in C-P B. Median OS: 19.6 months in ALBI-1, 10.5 months in ALBI-2, 3.1 months in ALBI-3.
Han Y. et al. (2021) [44]	Retrospective	REG+TACE after failure to SOR	Median OS: 14.3 months. Median PFS: 9.1 months. Median TTP: 9.1 months. The BCLC stage was not found to correlate with OS, PFS, and TTP in Cox regression analysis.
Lee I.-C. et al. (2022) [42]	Retrospective	108 patients BCLC B/C: 18.5%/81.5% ALBI 1/2/3: 40.7%/58.3%/0.9% C-P A/B: 84.3%/15.7% REG after SOR	***Better results in early switch from sorafenib to regorafenib*** Evaluation considering BCLC stage and liver function at the time of the switch (ALBI-1 vs. ALBI-2 vs. ALBI-3). No significant difference in PFS between BCLC-B and BCLC-C. Better OS in lower ALBI: 2–3/1 → HR 2.758, 95% CI: 1.458–5.216, *p* = 0.002. Better post-progression survival in lower ALBI: 2/1 → HR 4.499, 95% CI: 1.541–13.137, *p* = 0.006 and 3/1 → HR 26.926, 95% CI: 6.638–109.227, *p* < 0.001.
** *Lenvatinib* **			
Kudo M et al. (2019) [45]	Prospective	176 patients Lenvatinib vs. TACE in BCLC-B, C-P A No prior TACE/systemic treatment 30 patients in the lenvatinib group vs. 60 patients in the TACE group	***Lenvatinib superior to TACE*** Change in ALBI score during treatment: −2.61 to −2.61 in lenvatinib patients (*p* = 0.254) and from −2.66 to −2.09 in TACE patients (*p* < 0.01). ORR: 73.3% vs. 33.3%; *p* < 0.001. Median PFS: 16 vs. 3 months; *p* < 0.001. OS: 37.9 vs. 21.3 months; HR: 0.48, *p* < 0.01.
Kobayashi S. et al. (2022) [52]	Prospective	Lenvatinib vs. TACE (historical controls) in BCLC-B2, C-P: 5–6 31 patients	***Lenvatinib potentially superior to TACE*** Historical controls of TACE: OS and 1-year survival rates for substage B2 patients: 15.6–26.9 months and 59.2–75.5%, respectively. With lenvatinib: Median OS: 17.0 months. 1-year survival rate: 71.0%. 2-year survival rate: 32.3%. Median PFS: 10.4 months. 1-year PFS rate: 42%. ORR: 22.6% and DCR: 90.3%.
Patwala K. et al. (2022) [53]	Retrospective	155 patients (BCLC stage C (69.7%) and BCLC stage B (27.7%); C-P A/B/C: 78.8%/19.7%/1.5%) Efficacy of lenvatinib	***Better results in C-P A and BCLC-B stage*** Median OS in the entire population: 7.7 months. Median PFS in the entire population: 5.3 months. OS: C-P B/C vs. C-P A: 5.6 vs. 12.5 months; *p* < 0.01. In multivariate analysis: BCLC stage HR: 2.50, 95% CI: 1.40–4.45, *p* < 0.01. A higher baseline C-P score was an independent predictor of PFS (HR: 1.24, 95% CI: 1.01–1.52, *p* = 0.04).
Hiraoka A. et al. (2021) [54]	Retrospective	To determine whether the cause of liver cirrhosis (NAFLD vs. non-NAFLD) had any additional impact on OS and PFS of HCC patients treated with lenvatinib	Cox regression analysis: elevated ALT (≥30 U/L) (HR 1.247, *p* = 0.029), modified ALBI grade 2b (HR 1.236, *p* = 0.047), elevated AFP (≥400 ng/mL) (HR 1.294, *p* = 0.014), and NAFLD (HR 0.763, *p* = 0.036) were significant prognostic factors of PFS. Higher AFP (≥400 ng/mL) (HR 1.402, *p* = 0.009), BCLC-C stage (HR 1.297, *p* = 0.035), later introduction of lenvatinib (HR 0.737, *p* = 0.014), and modified ALBI grade 2b (HR 1.875, *p* < 0.001) were independently associated with OS.
Ando Y. et al. (2021) [55]	Prospective	88 BCLC-B subjects previously controlled by taking lenvatinib Two groups: 30 patients continued with lenvatinib plus TACE 58 patients remained on lenvatinib BCLC stage B (100%); beyond up-to-seven criteria (68.4% vs. 63.16%); ALBI grade 2 (31.58% vs. 31.58%)	***Lenvatinib/TACE potentially superior to lenvatinib*** ORR: 63.2% vs. 63.2%. Multivariate analysis: addition of TACE (HR: 0.264, 95% CI: 0.087–0.802, *p* = 0.019) and C-P score 5 (HR: 0.223, 95% CI: 0.070–0.704, *p* = 0.011) were independent predictors of PFS. Median PFS: 11.6 vs. 10.1 months CR: 15.8% vs. 10.5%.
Fu Z. et al. (2021) [56]	Retrospective	LEN/TACE vs. TACE	***Lenvatinib/TACE superior to TACE*** 1-year OS: 88.4% vs. 79.2%. 2-year OS: 79.8% vs. 49.2%. 1-year PFS: 78.4% vs. 64.7 (*p* < 0.001). 2-year PFS rate: 45.5% vs. 38.0% (*p* < 0.001). ORR: 68.3% vs. 31.7% (*p* < 0.001). In BCLC-B patients: ORR: 69.7% vs. 38.5%, *p* = 0.016. PFS: HR: 0.149; 95% CI: 0.059–0.379, *p* < 0.001. OS: HR: 0.28; 95% CI: 0.092–0.853, *p* = 0.07.

HCC, hepatocellular carcinoma; RCT, randomized controlled trial; C-P, Child–Pugh; BCLC, Barcelona Clinic Liver Cancer; SOR, sorafenib; REG, regorafenib; TACE, trans-arterial chemoembolization; LEN, lenvatinib; TKIs, tyrosine kinase inhibitors; ORR, overall response rate; CR, complete response; PFS, progression-free survival; OS, overall survival; TTP, time to progression; ALT, alanine transferase; AFP, alpha-fetoprotein; ALBI, albumin–bilirubin; HR, hazard ratio; OR, odds ratio; DCR, disease control rate; AE, adverse events; CI, confidence interval.

**Table 2 cancers-16-00051-t002:** Efficacy of ICIs on intermediate (BCLC-B) HCC.

		*Immune Checkpoint Inhibitors (ICIs)*	
Authors/Year/Ref No.	Study	Description	Results
Hiraoka A. et al. (2022) [62]	Prospective	171 HCC patients treated with atezolizumab plus bevacizumab. A number of patients had BCLC-B HCCs (n = 68, 40%)	ORR in 6 weeks of 10.6%. DCR in 6 weeks of 79.6%. The study did not provide comparative data regarding ORR and DCR between BCLC-B and BCLC-C patients.
Hiraoka A. et al. (2023) [63]	Prospective	Atezolizumab/bevacizumab vs. lenvatinib (BCLC-B/C)	***Atezolizumab/bevacizumab potentially superior to lenvatinib*** No significant differences in response rates of the two groups by using the m-RECIST criteria. PFS and OS were comparable between the two groups. After adjusting with inverse probability weighting, the Atez/Bev group showed better PFS (0.5-/1-/1.5-years: 56.6%/31.6%/non-estimable vs. 48.6%/20.4%/11.2%, *p* < 0.0001) and OS (0.5-/1-/1.5-years: 89.6%/67.2%/58.1% vs. 77.8%/66.2%/52.7%, *p* = 0.002).
Yang X. et al. (2023) [64]	Retrospective	Regorafenib plus ICIs and TACE (R+ICIs+TACE) vs. regorafenib plus ICIs (R+ICIs), as second-line treatment for patients with HCC	***R/ICIs/TACE superior to R/TACE*** After propensity score matching: ORR: 34.8% vs. 4.3%, *p* = 0.009. PFS: 5.8 vs. 2.6 months, *p* = 0.014. Cox regression model: BCLC stage was not found to correlate with PFS or OS.
Liu et al. (2023) [65]	Meta-analysis	5 studies, 425 patients TACE plus TKIs plus ICIs vs. TACE plus TKIs	***ICIs/TKIs/TACE superior to TKIs/TACE*** Triple treatment → ORR (RR 1.53, 95% CI: 1.27–1.85, *p* < 0.01), PFS→ mean difference: 4.51 months, *p* < 0.01 OS→ mean difference: 5.75 months, *p* < 0.01. Fixed-effects models: OS among C-P A patients, considering treatment options: TACE/TKIs/ICIs vs. TACE/TKIs: 22.6 vs. 15.1 months, *p* = 0.05. OS among C-P B subjects: TACE/TKIs/ICIs and TACE/TKIs: 12.9 vs. 6.5 months, *p* < 0.01.

HCC, hepatocellular carcinoma; BCLC: Barcelona Clinic Liver Cancer; ICIs, immune checkpoint inhibitors; R, regorafenib; TACE, trans-arterial chemoembolization; TKIs, tyrosine kinase inhibitors; m-RECIST, modified Response Evaluation Criteria in Solid Tumors; ORR, overall response rate; DCR, disease control rate; PFS, progression-free survival; OS, overall survival; Atez, atezolizumab; Bev, bevacizumab; HR, hazard ratio; OR, odds ratio; CI, confidence interval.

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
