# Peer review of "Systemic Treatment in Intermediate Stage (Barcelona Clinic Liver Cancer-B) Hepatocellular Carcinoma"

_cancers, 2023, doi:10.3390/cancers16010051_

Round 1
Reviewer 1 Report
Comments and Suggestions for Authors
The review “Systematic treatment in intermediate stage (BCLC-B) hepatocellular carcinoma” summarizes the latest insights of pharmacological treatment options patients with BCLC-B stage tumors.
Overall, the article is of high relevance and excellently outlines in a very factual and comprehensible way the need of more effective treatment strategies for the intermediate stage HCC. If TACE is insufficient, the management should rapidly change to immune checkpoint inhibitors (ICIs) or tyrosine kinase inhibitors (TKIs) in order to increase the progression free and overall survival.
Only a few minor issues were found as seen below.
Comments:
1. It is recommended to add a short paragraph to the introduction briefly defining BCLC-B stage HCC.
2. The comma placement needs to be double-checked.
3. Please rotate both tables 1 and 2 by 90° for a more appealing style. Stretch especially columns “Author/…”, “Study” and “Description”.
4. Line 28: “850,000 new cases yearly [1].” Please cite the latest case numbers from 2020
5. Line 65: “…resulting to…” should be corrected to “…resulting in…”.
6. Line 154: Which C-P patients C-P A, B or C were excluded? Please clarify in the text.
7. Line 237: Spelling error; correct “Lenvatiinib” to “Lenvatinib”
8. Page 14: Figure 1 is appropriate but not very attractive.
Comments on the Quality of English LanguagePlease double-check comma placements (e.g. before "respectively").
Author Response
We would like to thank you for giving us the opportunity to submit a revised version of our manuscript. We appreciate the time and effort that you and the reviewers dedicated to providing feedback on our manuscript and are grateful for the insightful comments and valuable improvements to our paper. We believe that after completion of the suggested edits, the revised manuscript has benefitted from an improvement in overall presentation and clarity. To this end, we have incorporated the suggestions made by the reviewer to the best extent possible. In our revised manuscript, all revisions have been highlighted. Below you will find a point-by-point response to the reviewers’ comments, which appear in italics. All page numbers refer to the revised manuscript.
Reviewer 1:
“It is recommended to add a short paragraph to the introduction briefly defining BCLC-B stage HCC”.
Reply: We thank the reviewer for his/her recommendation. However, as other reviewers asked us to reduce the length of the introduction, it was very difficult to add a whole paragraph in order to describe the definition of BCLC-B. Hence, we decided to add a sentence instead of a whole paragraph. (page 2, lines 71-73). We also added the definition of BCLC-B in figure 1.
“The comma placement needs to be double-checked”.
Reply: We double-checked the comma placement, according to reviewer’s comment
“Please rotate both tables 1 and 2 by 90° for a more appealing style. Stretch especially columns “Author/…”, “Study” and “Description”.
Reply: We tried to rotate both tables 1 and 2, but as the two tables are very big, the result was disappointing. We apologize for that.
“Line 28: “850,000 new cases yearly [1].” Please cite the latest case numbers from 2020”
Reply: We cited the latest case numbers from 2000 (page2, line 27) and we substituted reference number 1 (page 14, lines 517-518).
“Line 65: “…resulting to…” should be corrected to “…resulting in…”.”
Reply: We corrected the mistake (page2, line 53).
“Line 154: Which C-P patients C-P A, B or C were excluded? Please clarify in the text”.
Reply: We clarified in the text that patients with C-P C and C-P B/8 were excluded (page 4, lines 145-145).
“Line 237: Spelling error; correct “Lenvatiinib” to “Lenvatinib””.
Reply: We corrected the spelling error (page 5, line 228).
“Page 14: Figure 1 is appropriate but not very attractive.”
Reply: We did not remove figure 1 as it is necessary. However, we tried to fix it in order to become more attractive.
Reviewer 2 Report
Comments and Suggestions for Authors
Manuscript ID cancers-2748293, entitled “Systematic treatment in intermediate stage (BCLC-B) hepatocellular carcinoma”.
Thank you for the opportunity to review this original article.
The study deals with topic of interest and a question that, although not new, it is still under investigation and pertinent to our clinical practice.
The Author, reviewed recent data on systemic treatment on BCLC-B HCCs, aiming to emphasize its potential role on management of BCLC-B patients.
Despite reports have shown that surgical resection could provide a safe and effective treatment also for intermediate-stage HCC (BCLC-B), the author did not mention this option (as well as the role of liver transplantation) in the entire manuscript. In my experience, hepatic resection for intermediate-stage HCC shows acceptable results in terms of perioperative morbidity and mortality, with better oncological outcomes in patients with lower number of lesions despite of their size (DOI: 10.1007/s13304-019-00649-w).
Few points have the potential to be improved:
1- The term systematic should be change with the term systemic.
2- The introduction should be shortened and appears not impactful.
In addition, I believe the concept of therapeutic hierarchy for patients with hepatocellular carcinoma should be better underlined in the introduction.
3- Please be more specific about eligible study designs and report the full search syntax more clearly.
4- The discussion is too long and would benefit from being streamlined and shortened.
Comments on the Quality of English LanguageExtensive editing of english language is required.
Author Response
We would like to thank you for giving us the opportunity to submit a revised version of our manuscript. We appreciate the time and effort that you and the reviewers dedicated to providing feedback on our manuscript and are grateful for the insightful comments and valuable improvements to our paper. We believe that after completion of the suggested edits, the revised manuscript has benefitted from an improvement in overall presentation and clarity. To this end, we have incorporated the suggestions made by the reviewer to the best extent possible. In our revised manuscript, all revisions have been highlighted. Below you will find a point-by-point response to the reviewers’ comments, which appear in italics. All page numbers refer to the revised manuscript.
Reviewer 2:
“Despite reports have shown that surgical resection could provide a safe and effective treatment also for intermediate-stage HCC (BCLC-B), the author did not mention this option (as well as the role of liver transplantation) in the entire manuscript. In my experience, hepatic resection for intermediate-stage HCC shows acceptable results in terms of perioperative morbidity and mortality, with better oncological outcomes in patients with lower number of lesions despite of their size (DOI: 10.1007/s13304-019-00649-w).”
Reply: We agree with the reviewer’s statement. However, the discussion of liver transplantation and surgical resection, as therapeutic options for BCLC-B HCCs, was out of the scope of this paper. The objective of this review was to designate new perspectives on management of BCLC-B HCCs by using systemic treatment. In order to make that clear, we added a paragraph in the discussion (page 10, lines 471-476). In addition, we changed the last paragraph in introduction (page 3, lines 87-94).
Few points have the potential to be improved:
“The term systematic should be change with the term systemic”.
Reply: We changed the term systematic with the term systemic
“The introduction should be shortened and appears not impactful.”
Reply: We shortened the introduction according to reviewer’s comment.
In addition, I believe the concept of therapeutic hierarchy for patients with hepatocellular carcinoma should be better underlined in the introduction.
Reply: We made changes in page 2 in order to make clear the hierarchy of systemic treatments in BCLC-B stages (page 2, lines 51-71).
“Please be more specific about eligible study designs and report the full search syntax more clearly”.
Reply: In order to clarify this issue, we added a paragraph at the end of introduction (page 3, lines 87-94).
“The discussion is too long and would benefit from being streamlined and shortened”.
Reply: We shortened the discussion according to reviewer’s comment
Comments on the Quality of English Language
“Extensive editing of english language is required”.
Reply: We corrected the grammar and syntax errors.
Round 2
Reviewer 2 Report
Comments and Suggestions for Authors
no other comments
Comments on the Quality of English Languageno other comments